# FRAGE: Frequency-Agnostic Word Representation

**Chengyue Gong**[1]
cygong@pku.edu.cn

**Di He**[2]
di_he@pku.edu.cn

**Xu Tan**[3]
xu.tan@microsoft.com

**Tao Qin**[3]
taoqin@microsoft.com

**Liwei Wang**[2,4]
wanglw@cis.pku.edu.cn

**Tie-Yan Liu**[3]
tie-yan.liu@microsoft.com

[1]Peking University
[2]Key Laboratory of Machine Perception, MOE, School of EECS, Peking University
[3]Microsoft Research Asia
[4]Center for Data Science, Peking University, Beijing Institute of Big Data Research

## Abstract

Continuous word representation (aka word embedding) is a basic building block in many neural network-based models used in natural language processing tasks. Although it is widely accepted that words with similar semantics should be close to each other in the embedding space, we find that word embeddings learned in several tasks are biased towards word frequency: the embeddings of high-frequency and low-frequency words lie in different subregions of the embedding space, and the embedding of a rare word and a popular word can be far from each other even if they are semantically similar. This makes learned word embeddings ineffective, especially for rare words, and consequently limits the performance of these neural network models. In this paper, we develop *FRequency-AGnostic word Embedding* (FRAGE) which is a neat, simple yet effective way to learn word representation using adversarial training. We conducted comprehensive studies on ten datasets across four natural language processing tasks, including word similarity, language modeling, machine translation, and text classification. Results show that with FRAGE, we achieve higher performance than the baselines in all tasks.

## 1  Introduction

Word embeddings, which are distributed and continuous vector representations for word tokens, have been one of the basic building blocks for many neural network-based models used in natural language processing (NLP) tasks, such as language modeling [18, 16], text classification [24, 7] and machine translation [4, 5, 40, 38, 11]. Different from classic one-hot representation, the learned word embeddings contain semantic information which can measure the semantic similarity between words [28], and can also be transferred into other learning tasks [29, 3].

In deep learning approaches for NLP tasks, word embeddings act as the inputs of the neural network and are usually trained together with neural network parameters. As the inputs of the neural network, word embeddings carry all the information of words that will be further processed by the network, and the quality of embeddings is critical and highly impacts the final performance of the learning task [15]. Unfortunately, we find the word embeddings learned by many deep learning approaches are far from perfect. As shown in Figure 1(a) and 1(b), in the embedding space learned by word2vec model, the nearest neighbors of word "Peking" includes "quickest", "multicellular", and "epigenetic", which

are not semantically similar, while semantically related words such as "Beijing" and "China" are far from it. Similar phenomena are observed from the word embeddings learned from translation tasks.

With a careful study, we find a more general problem which is rooted in low-frequency words in the text corpus. Without any confusion, we also call high-frequency words as popular words and call low-frequency words as rare words. As is well known [23], the frequency distribution of words roughly follows a simple mathematical form known as Zipf's law. When the size of a text corpus grows, the frequency of rare words is much smaller than popular words while the number of unique rare words is much larger than popular words. Interestingly, the learned embeddings of rare words and popular words behave differently. (1) In the embedding space, a popular word usually has semantically related neighbors, while a rare word usually does not. Moreover, the nearest neighbors of more than 85% rare words are rare words. (2) Word embeddings $encode$ frequency information. As shown in Figure 1(a) and 1(b), the embeddings of rare words and popular words actually lie in different subregions of the space. Such a phenomenon is also observed in [29].

We argue that the different behaviors of the embeddings of popular words and rare words are problematic. First, such embeddings will affect the semantic understanding of words. We observe more than half of the rare words are nouns or variants of popular words. Those rare words should have similar meanings or share the same topics with popular words. Second, the neighbors of a large number of rare words are semantically unrelated rare words. To some extent, those word embeddings encode more frequency information than semantic information which is not good from the view of semantic understanding. It will consequently limit the performance of down-stream tasks using the embeddings. For example, in text classification, it cannot be well guaranteed that the label of a sentence does not change when you replace one popular/rare word in the sentence by its rare/popular alternatives.

To address this problem, in this paper, we propose an adversarial training method to learn *FRequency-AGnostic word Embedding* (FRAGE). For a given NLP task, in addition to minimizing the task-specific loss by optimizing the task-specific parameters together with word embeddings, we introduce another discriminator, which takes a word embedding as input and classifies whether it is a popular/rare word. The discriminator optimizes its parameters to maximize its classification accuracy, while word embeddings are optimized towards a low task-dependent loss as well as fooling the discriminator to misclassify the popular and rare words. When the whole training process converges and the system achieves an equilibrium, the discriminator cannot well differentiate popular words from rare words. Consequently, rare words lie in the same region as and are mixed with popular words in the embedding space. Then FRAGE will catch better semantic information and help the task-specific model to perform better.

We conduct experiments on four types of NLP tasks, including three word similarity tasks, two language modeling tasks, three sentiment classification tasks, and two machine translation tasks to test our method. In all tasks, FRAGE outperforms the baselines. Specifically, in language modeling and machine translation, we achieve better performance than the state-of-the-art results on PTB, WT2 and WMT14 English-German datasets.

## 2 Background

### 2.1 Word Representation

Words are the basic units of natural languages, and distributed word representations (i.e., word embeddings) are the basic units of many models in NLP tasks including language modeling [18, 16] and machine translation [4, 5, 40, 38, 11]. It has been demonstrated that word representations learned from one task can be transferred to other tasks and achieve competitive performance [3].

While word embeddings play an important role in neural network-based models in NLP and achieve great success, one technical challenge is that the embeddings of rare words are difficult to train due to their low frequency of occurrences. [35] develops a novel way to split each word into sub-word units which is widely used in neural machine translation. However, the low-frequency sub-word units are still difficult to train: [31] provides a comprehensive study which shows that the rare (sub)words are usually under-estimated in neural machine translation: during inference step, the model tends to choose popular words over their rare alternatives.

## 2.2 Adversarial Training

The basic idea of our work to address the above problem is adversarial training, in which two or more models learn together by pursuing competing goals. A representative example of adversarial training is Generative Adversarial Networks (GANs) [13, 34] for image generation [33, 42, 2], in which a discriminator and a generator compete with each other: the generator aims to generate images similar to the natural ones, and the discriminator aims to detect the generated ones from the natural ones. Recently, adversarial training has been successfully applied to NLP tasks [6, 22, 21]. [6, 22] introduce an additional discriminator to differentiate the semantics learned from different languages in non-parallel bilingual data. [21] develops a discriminator to classify whether a sentence is created by human or generated by a model.

Our proposed method is under the adversarial training framework but not exactly the conventional generator-discriminator approach since there is no generator in our scenario. For an NLP task and its neural network model (including word embeddings), we introduce a discriminator to differentiate embeddings of popular words and rare words; while the NN model aims to fool the discriminator and minimize the task-specific loss simultaneously.

Our work is also weakly related to adversarial domain adaptation which attempts to mitigate the negative effects of domain shift between training and testing [9, 36]. The difference between this work and adversarial domain adaptation is that we do not target at the mismatch between training and testing; instead, we aim to improve the effectiveness of word embeddings and consequently improve the performance of end-to-end NLP tasks.

## 3 Empirical Study

In this section, we study the embeddings of popular words and rare words based on the models trained from Google News corpora using word2vec [1] and trained from WMT14 English-German translation task using Transformer [38]. The implementation details can be found in [12].

**Experimental Design**   In both tasks, we simply set the top $20\%$ frequent words in vocabulary as popular words and denote the rest as rare words (roughly speaking, we set a word as a rare word if its relative frequency is lower than $10^{-6}$ in WMT14 dataset and $10^{-7}$ in Google News dataset). We have tried other thresholds such as $10\%$ or $25\%$ and found the observations are similar.

We study whether the semantic relationship between two words is reasonable. To achieve this, we randomly sampled some rare/popular words and checked the embeddings trained from different tasks. For each sampled word, we determined its nearest neighbors based on the cosine similarity between its embeddings and others'.[2] We also manually chose words which are semantically similar to it. For simplicity, for each word, we call the nearest words predicted from the embeddings as *model-predicted neighbors*, and call our chosen words as *semantic neighbors*.

**Observation**   To visualize word embeddings, we reduce their dimensionalities by SVD and plot two cases in Figure 1. More cases and other studies without dimensionality reduction can be found in Section 5.

We find that the embeddings trained from different tasks share some common patterns. For both tasks, more than 90% of model-predicted neighbors of rare words are rare words. For each rare word, the model-predicted neighbor is usually not semantically related to this word, and semantic neighbors we chose are far away from it in the embedding space. In contrast, the model-predicted neighbors of popular words are very reasonable.

As the patterns in rare words are different from that of popular words, we further check the whole embedding matrix to make a general understanding. We also visualize the word embeddings using SVD by keeping the two directions with top-2 largest eigenvalues as in [28, 30] and plot them in Figure 1(c) and 1(d). From the figure, we can see that the embeddings actually *encode* frequencies to a certain degree: the rare words and popular words lie in different regions after this linear projection,

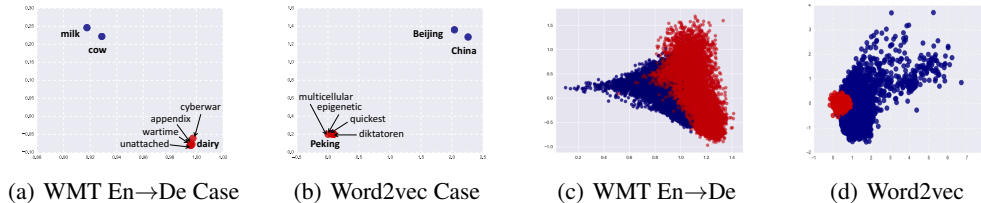

| (a) WMT En→De Case | (b) Word2vec Case | (c) WMT En→De | (d) Word2vec |

Figure 1: Case study of the embeddings trained from WMT14 translation task using Transformer and trained from Google News dataset using word2vec is shown in (a) and (b). (c) and (d) show the visualization of embeddings trained from WMT14 translation task using Transformer and trained from Google News dataset using word2vec. Red points represent rare words and blue points represent popular words. In (a) and (b), we highlight the semantic neighbors in bold.

and thus they occupy different regions in the original embedding space. This strange phenomenon is also observed in other learned embeddings (e.g.CBOW and GLOVE) and mentioned in [30].

**Explanation** From the empirical study above, we can see that the occupied spaces of popular words and rare words are different and here we intuitively explain a possible reason. We simply take word2vec as an example which is trained by stochastic gradient descent. During training, the sample rate of a popular word is high and the embedding of a popular word updates frequently. For a rare word, the sample rate is low and its embedding rarely updates. According to our study, on average, the moving distance of the embedding for a popular word is twice longer than that of a rare word during training. As all word embeddings are usually initialized around the origin with a small variance, we observe in the final model, the embeddings of rare words are still around the origin and the popular words have moved far away.

**Discussion** We have strong evidence that the current phenomena are problematic. First, according to our study,[3] in both tasks, more than half of the rare words are nouns, e.g., company names, city names. They may share some similar topics to popular entities, e.g., big companies and cities; around 10% percent of rare words include a hyphen (which is usually used to join popular words), and over 30% rare words are different PoSs of popular words. These words should have mixed or similar semantics to some popular words. These facts show that rare words and popular words should lie in the same region of the embedding space, which is different from what we observed. Second, as we can see from the cases, for rare words, model-predicted neighbors are usually not semantically related words but frequency-related words (rare words). This shows, for rare words, the embeddings encode more frequency information than semantic information. It is not good to use such word embeddings into semantic understanding tasks, e.g., text classification, language modeling, language understanding, and translation.

## 4    Our Method

In this section, we present our method to improve word representations. As we have a strong prior that many rare words should share the same region in the embedding space as popular words, the basic idea of our algorithm is to train the word embeddings in an adversarial framework: We introduce a discriminator to categorize word embeddings into two classes: popular ones or rare ones. We hope the learned word embeddings not only minimize the task-specific training loss but also fool the discriminator. By doing so, the frequency information is removed from the embedding and we call our method frequency-agnostic word embedding (FRAGE).

We first define some notations and then introduce our algorithm. We develop three types of notations: embeddings, task-specific parameters/loss, and discriminator parameters/loss.

Denote $\theta^{emb} \in R^{d \times |V|}$ as the word embedding matrix to be learned, where $d$ is the dimension of the embedding vectors and $|V|$ is the vocabulary size. Let $V_{pop}$ denote the set of popular words and $V_{rare} = V \setminus V_{pop}$ denote the set of rare words. Then the embedding matrix $\theta^{emb}$ can be divided

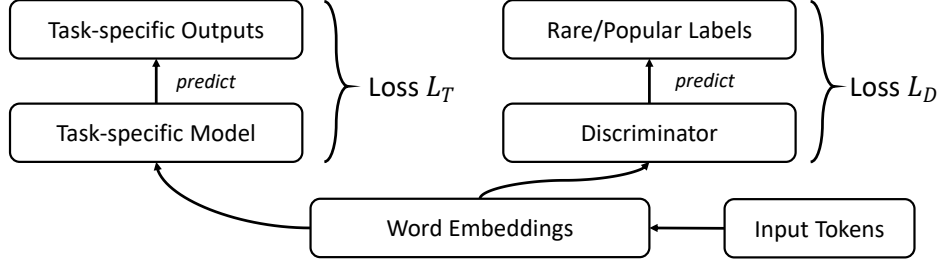

Figure 2: The proposed learning framework includes a task-specific predictor and a discriminator, whose function is to classify rare and popular words. Both modules use word embeddings as the input.

into two parts: $\theta^{emb}_{pop}$ for popular words and $\theta^{emb}_{rare}$ for rare words. Let $\theta^{emb}_w$ denote the embedding of word $w$. Let $\theta^{model}$ denote all the other task-specific parameters except word embeddings. For instance, for language modeling, $\theta^{model}$ is the parameters of the RNN or LSTM; for neural machine translation, $\theta^{model}$ is the parameters of the encoder, attention module, and decoder.

Let $L_T(S; \theta^{model}, \theta^{emb})$ denote the task-specific loss over a dataset $S$. Taking language modeling as an example, the loss $L_T(S; \theta^{model}, \theta^{emb})$ is defined as the negative log likelihood of the data:

$$L_T(S; \theta^{model}, \theta^{emb}) = -\frac{1}{|S|} \sum_{y \in S} \log P(y; \theta^{model}, \theta^{emb}), \tag{1}$$

where $y$ is a sentence.

Let $f_{\theta^D}$ denote a discriminator with parameters $\theta^D$, which takes a word embedding as input and outputs a confidence score between 0 and 1 indicating how likely the word is a rare word. Let $L_D(V; \theta^D, \theta^{emb})$ denote the loss of the discriminator:

$$L_D(V; \theta^D, \theta^{emb}) = \frac{1}{|V_{pop}|} \sum_{w \in V_{pop}} \log f_{\theta^D}(\theta^{emb}_w) + \frac{1}{|V_{rare}|} \sum_{w \in V_{rare}} \log(1 - f_{\theta^D}(\theta^{emb}_w)). \tag{2}$$

Following the principle of adversarial training, we develop a minimax objective to train the task-specific model ($\theta^{model}$ and $\theta^{emb}$) and the discriminator ($\theta^D$) as below:

$$\min_{\theta^{model}, \theta^{emb}} \max_{\theta^D} L_T(S; \theta^{model}, \theta^{emb}) - \lambda L_D(V; \theta^D, \theta^{emb}), \tag{3}$$

where $\lambda$ is a coefficient to trade off the two loss terms. We can see that when the model parameter $\theta^{model}$ and the embedding $\theta^{emb}$ are fixed, the optimization of the discriminator $\theta^D$ becomes

$$\max_{\theta^D} -\lambda L_D(V; \theta^D, \theta^{emb}), \tag{4}$$

which is to minimize the classification error of popular and rare words. When the discriminator $\theta^D$ is fixed, the optimization of $\theta^{model}$ and $\theta^{emb}$ becomes

$$\min_{\theta^{model}, \theta^{emb}} L_T(S; \theta^{model}, \theta^{emb}) - \lambda L_D(V; \theta^D, \theta^{emb}), \tag{5}$$

i.e., to optimize the task performance as well as fooling the discriminator. We train $\theta^{model}$, $\theta^{emb}$ and $\theta^D$ iteratively by stochastic gradient descent or its variants. The general training process is shown in Algorithm 1.

## 5 Experiment

We test our method on a wide range of tasks, including word similarity, language modeling, machine translation, and text classification. For each task, we choose the state-of-the-art architecture together with the state-of-the-art training method as our baseline [4].

**Algorithm 1** Proposed Algorithm
---
1: **Input**: Dataset $S$, vocabulary $V = V_{pop} \cup V_{rare}, \theta^{model}, \theta^{emb}, \theta^D$.
2: **repeat**
3:     Sample a minibatch $\hat{S}$ from $S$.
4:     Sample a minibatch $\hat{V} = \hat{V}_{pop} \cup \hat{V}_{rare}$ from $V$.
5:     Update $\theta^{model}, \theta^{emb}$ by gradient descent according to Eqn. (5) with data $\hat{S}$.
6:     Update $\theta^D$ by gradient ascent according to Eqn. (4) with vocabulary $\hat{V}$.
7: **until** Converge
8: **Output**: $\theta^{model}, \theta^{emb}, \theta^D$.
---

For fair comparisons, for each task, our method shares the same model architecture as the baseline. The only difference is that we use the original task-specific loss function with an additional adversarial loss as in Eqn. (3). Dataset description and hyper-parameter configurations can be found in [12].

## 5.1 Settings

We conduct experiments on the following tasks.

**Word Similarity** evaluates the performance of the learned word embeddings by calculating the word similarity: it evaluates whether the most similar words of a given word in the embedding space are consistent with the ground-truth, in terms of Spearman's rank correlation. We use the skip-gram model as our baseline model [28][5], and train the embeddings using Enwik9[6]. We test the baseline and our method on three datasets: RG65, WS, and RW. The RW dataset is a dataset for the evaluation of rare words. Following common practice [28, 1, 32, 29], we use cosine distance while computing the similarity between two word embeddings.

**Language Modeling** is a basic task in natural language processing. The goal is to predict the next word conditioned on previous words and the task is evaluated by perplexity. We do experiments on two widely used datasets [25, 26, 41], Penn Treebank (PTB) [27] and WikiText-2 (WT2) [26]. We choose two recent works as our baselines: the AWD-LSTM model[7] [25] and the AWD-LSTM-MoS model,[8] [41]. AWD-LSTM [25] is a weight-dropped LSTM which uses Drop Connect on hidden-to-hidden weights as a means of recurrent regularization. The model is trained by NT-ASGD, which is a variant of the averaged stochastic gradient method. The training process has two steps, in the second step, the model is finetuned using another configuration of NT-ASGD. AWD-LSTM-MoS [41] uses the *Mixture of Softmaxes* structure to the vanilla AWD-LSTM and achieves the state-of-the-art result on PTB and WT2.

**Machine Translation** is a popular task in both deep learning and natural language processing. We choose two datasets: WMT14 English-German and IWSLT14 German-English datasets, which are evaluated in terms of BLEU score[9]. We use Transformer [38] as the baseline model. Transformer [38] is a recently developed architecture in which the *self-attention network* is used during encoding and decoding step. It achieves the best performances on several machine translation tasks, e.g. WMT14 English-German, WMT14 English-French datasets. We use *transformer_base* and *transformer_big* configurations following `tensor2tensor` [37][10].

**Text Classification** is a conventional machine learning task and is evaluated by accuracy. Following the setting in [20], we implement a Recurrent CNN-based model[11] and test it on AG's news corpus (AGs), IMDB movie review dataset (IMDB) and 20 Newsgroups (20NG). RCNN [20] contains both

recurrent and convolutional layers to catch the key components in texts and is widely used in text classification tasks.

In all tasks, we simply set the top $20\%$ frequent words in vocabulary as popular words and denote the rest as rare words, which is the same as our empirical study. For all the tasks except training skip-gram model, we use full-batch gradient descent to update the discriminator. For training skip-gram model, mini-batch stochastic gradient descent is used to update the discriminator with a batch size 3000, since the vocabulary size is large. For language modeling and machine translation tasks, we use logistic regression as the discriminator. For other tasks, we find using a shallow neural network with one hidden layer is more efficient and we set the number of nodes in the hidden layer as 1.5 times embedding size. In all tasks, we set the hyper-parameter $\lambda$ to 0.1.

| RG65 | | WS | | RW | |
|---|---|---|---|---|---|
| Orig. | with FRAGE | Orig. | with FRAGE | Orig. | with FRAGE |
| 75.63 | **78.78** | 66.74 | **69.35** | 52.67 | **58.12** |

Table 1: Results on three word similarity datasets.

| | | **Paras** | **Orig.** | | **with FRAGE** | |
|---|---|---|---|---|---|---|
| | | | Validation | Test | Validation | Test |
| **PTB** | AWD-LSTM w/o finetune[25] | 24M | 60.7 | 58.8 | 60.2 | 58.0 |
| | AWD-LSTM[25] | 24M | 60.0 | 57.3 | 58.1 | 56.1 |
| | AWD-LSTM + continuous cache pointer[25] | 24M | 53.9 | 52.8 | **52.3** | **51.8** |
| | AWD-LSTM-MoS w/o finetune[41] | 24M | 58.08 | 55.97 | 57.55 | 55.23 |
| | AWD-LSTM-MoS[41] | 24M | 56.54 | 54.44 | 55.52 | 53.31 |
| | AWD-LSTM-MoS + dynamic evaluation[41] | 24M | 48.33 | 47.69 | **47.38** | **46.54** |
| **WT2** | AWD-LSTM w/o finetune[25] | 33M | 69.1 | 67.1 | 67.9 | 64.8 |
| | AWD-LSTM[25] | 33M | 68.6 | 65.8 | 66.5 | 63.4 |
| | AWD-LSTM + continuous cache pointer[25] | 33M | 53.8 | 52.0 | **51.0** | **49.3** |
| | AWD-LSTM-MoS w/o finetune[41] | 35M | 66.01 | 63.33 | 64.86 | 62.12 |
| | AWD-LSTM-MoS[41] | 35M | 63.88 | 61.45 | 62.68 | 59.73 |
| | AWD-LSTM-MoS + dynamic evaluation[41] | 35M | 42.41 | 40.68 | **40.85** | **39.14** |

Table 2: Perplexity on validation and test sets on Penn Treebank and WikiText2. Smaller the perplexity, better the result. Baseline results are obtained from [25, 41]. "Paras" denotes the number of model parameters.

## 5.2 Results

In this subsection, we provide the experimental results of all tasks. For simplicity, we use "with FRAGE" as our proposed method in the tables.

**Word Similarity** The results on three word similarity tasks are listed in Table 1. From the table, we can see that our method consistently outperforms the baseline on all datasets. In particular, we outperform the baseline for about 5.4 points on the rare word dataset RW. This result shows that our method improves the representation of words, especially the rare words.

**Language Modeling** The results of language modeling on PTB and WT2 datasets are presented in Table 2. We test our model and the baselines at several checkpoints used in the baseline papers: without finetune, with finetune, with post-process (continuous cache pointer [14] or dynamic evaluation [19]). In all these settings, our method outperforms the two baselines. On PTB dataset, our method improves

the AWD-LSTM and AWD-LSTM-MoS baseline by 0.8/1.2/1.0 and 0.76/1.13/1.15 points in test set at different checkpoints. On WT2 dataset, which contains more rare words, our method achieves larger improvements. We improve the results of AWD-LSTM and AWD-LSTM-MoS by 2.3/2.4/2.7 and 1.15/1.72/1.54 in terms of test perplexity, respectively.

| WMT En→De | | IWSLT De→En | |
|---|---|---|---|
| **Method** | **BLEU** | **Method** | **BLEU** |
| ByteNet[17] | 23.75 | DeepConv[10] | 30.04 |
| ConvS2S[11] | 25.16 | Dual transfer learning [39] | 32.35 |
| Transformer Base[38] | 27.30 | ConvS2S+SeqNLL [8] | 32.68 |
| Transformer Base with FRAGE | **28.36** | ConvS2S+Risk [8] | 32.93 |
| Transformer Big[38] | 28.40 | Transformer | 33.12 |
| Transformer Big with FRAGE | **29.11** | Transformer with FRAGE | **33.97** |

Table 3: BLEU scores on test set on WMT2014 English-German and IWSLT German-English tasks.

**Machine Translation** The results of neural machine translation on WMT14 English-German and IWSLT14 German-English tasks are shown in Table 3. We outperform the baselines for 1.06/0.71 in the term of BLEU in *transformer_base* and *transformer_big* settings in WMT14 English-German task, respectively. The model learned from adversarial training also outperforms the original one in IWSLT14 German-English task by 0.85. These results show improving word embeddings can achieve better results in more complicated tasks and larger datasets.

| AG's | | IMDB | | 20NG | |
|---|---|---|---|---|---|
| **Orig.** | **with FRAGE** | **Orig.** | **with FRAGE** | **Orig.** | **with FRAGE** |
| 90.47% | **91.73%** | 92.41% | **93.07%** | 96.49%[20] | **96.93%** |

Table 4: Accuracy on test sets of AG's news corpus (AG's), IMDB movie review dataset (IMDB) and 20 Newsgroups (20NG) for text classification.

**Text Classification** The results are listed in Table 4. Our method outperforms the baseline method for 1.26%/0.66%/0.44% on three different datasets.

As a summary, our experiments on four different tasks with 10 datasets verify the effectiveness of our method. We provide case study and qualitative analysis of the model with and without our method in Table 5 and Figure 3. By comparing the cases, we find that, with our method, the word similarities are improved and popular/rare words are better mixed together. More cases are shown in [12].

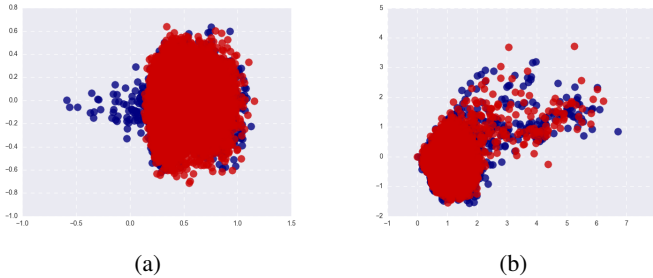

(a)&emsp;&emsp;&emsp;&emsp;&emsp;&emsp;&emsp;&emsp;(b)

Figure 3: These figures show that, in different tasks, the embeddings of rare and popular words are better mixed together after applying our method.

| Orig. | | Orig. | |
|---|---|---|---|
| **Word: citizens** | **Word: citizenship*** | **Word: accepts*** | **Word: bacterial*** |
| **Model-predicted neighbor** | | | |
| clinicians* | bliss* | announces* | multicellular* |
| astronomers* | pakistanis* | digs* | epigenetic* |
| westliche | dismiss* | externally* | isotopic* |
| adults | reinforces* | empowers* | conformational* |
| **Semantic neighbor + Model-predicted Ranking** | | | |
| citizen*:771 | citizen*:10745 | accepted*:21109 | bacteria*:116 |
| citizenship*:832 | citizens:11706 | accept:30612 | chemical:233 |
| **Orig. with FRAGE** | | **Orig. with FRAGE** | |
| **Word: citizens** | **Word: citizenship*** | **Word: accepts*** | **Word: bacterial*** |
| **Model-predicted neighbor** | | | |
| homes | population | registered | myeloproliferative* |
| citizen* | städtischen* | tolerate* | metabolic* |
| bürger | dignity | recognizing* | bacteria* |
| population | bürger | accepting* | apoptotic* |
| **Semantic neighbor + Model-predicted Ranking** | | | |
| citizen*:2 | citizen*:79 | accepted*:26 | bacteria* : 3 |
| citizenship*:40 | citizens:7 | accept:29 | chemical: 8 |

Table 5: Case study for the original model and our method. Rare words are marked by "*". For each word, we list its model-predicted neighbors. Moreover, we also show the ranking positions of the semantic neighbors based on cosine similarity. As we can see, the ranking positions of the semantic neighbors are very low for the original model.

# 6 Conclusion

In this paper, we find that word embeddings learned in several tasks are biased towards word frequency: the embeddings of high-frequency and low-frequency words lie in different subregions of the embedding space. This makes learned word embeddings ineffective, especially for rare words, and consequently limits the performance of these neural network models. We propose a neat, simple yet effective adversarial training method to improve the model performance which is verified in a wide range of tasks.

We will explore several directions in the future. First, we will investigate the theoretical aspects of word embedding learning and our adversarial training method. Second, we will study more applications which have the similar problem even beyond NLP.

# Acknowledgement

This work is supported by National Basic Research Program of China (973 Program) (grant no. 2015CB352502), NSFC (61573026) and BJNSF (L172037) and a grant from Microsoft Research Asia. We would like to thank the anonymous reviewers for their valuable comments on our paper.

## Footnotes

[1]https://code.google.com/archive/p/word2vec/

[2]Cosine distance is the most popularly used metric in literature to measure semantic similarity [28, 32, 29]. We also have tried other metrics, e.g., Euclid distance, and the phenomena still exist.

[3]We use the POS tagger from Natural Language Toolkit, https://github.com/nltk.

[4]Code for our implementation is available at https://github.com/ChengyueGongR/FrequencyAgnostic

[5]https://github.com/tensorflow/models/blob/master/tutorials/embedding

[6]http://mattmahoney.net/dc/textdata.html

[7]https://github.com/salesforce/awd-lstm-lm

[8]https://github.com/zihangdai/mos

[9]https://github.com/moses-smt/mosesdecoder/blob/master/scripts/generic/multi-bleu.perl

[10]To improve the training for imbalanced labeled data, a common method is to adjust loss function by reweighting the training samples; To regularize the parameter space, a common method is to use $l_2$ regularization. We tested these methods in machine translation and found the performance is not good. Detailed analysis is provided in [12]

[11]https://github.com/brightmart/text_classification

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
