[Reviews · NeurIPS 2018]

Reviewer 1



It's been known that word embeddings trained by popular methods (e.g. word2vec) show a strong bias towards popular words, and that the frequency information is strongly encoded in the embedding vector at the expense of semantic similarity of rare words. This paper is an attempt to alleviate this issue using adversarial training. The authors borrow the DANN architecture to train 'frequency agnostic' word embeddings. The resulting embeddings are shown to better encode semantically similar rare words and improve performance on downstream tasks (word similarity, LM training and machine translation). The paper is clearly written. Both the qualitative and quantitative evaluations are of high quality. I'm convinced that the method really works to improve rare word representations. It is also a noval application of adversarial training and has wide applicability. One question I still have which is unanswered in the paper is this: - It's clear that the adversarial loss mixes the frequent and rare words together in the embedding space. However, it is not clear why this method results in semantically similar rare words coming closer together. There is nothing in the method that does this directly. It could conceivably be the case that rare words are mixed together with popular words, while still being far from their semantic neighbors. If the "why" question was better answered, I would be much more confident in my rating. Other: it would be good to see fig 1 and the full table 7 in supplementary material be reproduced with the new model

Reviewer 2



*Summary: This paper presents a simple regularization technique for word embeddings. The core idea is adding an adversarial loss which is recently widely used in many NLP papers as mentioned in this paper submission. The authors define two frequency-based domains: major words and rare words. The proposed approach is easy to use and seems helpful in improving accuracy of several NLP tasks. *Strengths: - The frequency-based analysis on word embeddings is interesting and motivates this work well. - The method is easy to follow, although we have to additionally tune a few hyperparameters (the frequency threshold and the coefficient for the additional loss). - The method leads to consistent improvements upon the baselines on the NLP tasks used in their experiments. *Weaknesses: - Figure 1 and 2 well motive this work, but in the main body of this paper I cannot see what happens to these figures after applying the proposed adversarial training. It is better to put together the images before and after applying your method in the same place. Figure 2 does not say anything about details (we can understand the very brief overview of the positions of the embeddings), and thus these figures could be smaller for better space usage. - For the LM and NMT models, did you use the technique to share word embedding and output softmax matrices as in [1]? The transformer model would do this, if the transformer implementations are based on the original paper. If so, your method affects not only the input word embeddings, but also the output softmax matrix, which is not a trivial side effect. This important point seems missing and not discussed. If the technique is not used, the strength of the proposed method is not fully realized, because the output word embeddings could still capture simple frequency information. [1] Inan et al., Tying Word Vectors and Word Classifiers: A Loss Framework for Language Modeling, ICLR 2017. - There are no significance test or discussions about the significance of the score differences. - It is not clear how the BLEU improvement comes from the proposed method. Did you inspect whether rare words are more actively selected in the translations? Otherwise, it is not clear whether the expectations of the authors actually happened. - Line 131: The authors mention standard word embeddings like word2vec-based and glove-based embeddings, but recently subword-based embeddings are also used. For example, fasttex embeddings are aware of internal character n-gram information, which is helpful in capturing information about rare words. By inspecting the character n-grams, it is sometimes easy to understand rare words' brief properties. For example, in the case of "Peking", we can see the words start from a uppercase character and ends by the suffix "ing", etc. It makes this paper more solid to compare the proposed method with such character n-gram-based methods [2, 3]. [2] Bojanowski et al., Enriching Word Vectors with Subword Information, TACL. [3] Hashimoto et al., A Joint Many-Task Model: Growing a Neural Network for Multiple NLP Tasks, EMNLP 2017. *Minor comments: - Line 21: I think the statement "Different from classic one-hot representation" is not necessary, because anyway word embeddings are still based on such one-hot representations (i.e., the word indices). An embedding matrix is just a weight matrix for the one-hot representations. - Line 29: Word2vec is not a model, but a toolkit which implements Skipgram and CBOW with a few training options. - The results on Table 6 in the supplementary material could be enough to be tested on the dev set. Otherwise, there are too many test set results. * Additional comments after reading the author response Thank you for your kind reply to my comments and questions. I believe that the draft will be further improved in the camera-ready version. One additional suggestion is that the title seems to be too general. The term "adversarial training" has a wide range of meanings, so it would be better to include your contribution in the title; for example, "Improving Word Embeddings by Frequency-based Adversarial Training" or something.

Reviewer 3



The authors propose a learning framework to improve the word embedding. They start from studying a common phenomenon in popular word embedding models such as word2vec and Glove model, i.e., the rare words usually do not have semantically related neighbors because of their low frequency in the corpus. This phenomenon is also reported in the paper "All-but-the-top: Simple and effective post processing for word representations(ICLR 2018)." However, this paper (ICLR 2018) simply eliminates the common mean vector of all word vectors and a few top dominating directions from the word embedding. To address this problem, the authors introduce a discriminator in the learning framework to categorize word embedding into two classes: rare words or popular words. For each specific NLP task, this learning framework includes two parts: a task-specific model and a discriminator. The learned word embedding not only minimizes the task-specific training loss but also fools the discriminator. The experimental results demonstrate their framework. It’s not the first time that people introduce the adversarial training into NLP tasks. In "Professor forcing: A new algorithm for training recurrent networks(NIPS 2016)" and "Word translation without parallel data(ICLR 2018)", the authors both adopted the adversarial training, but from different aspects. This paper mainly focuses on the word embedding, which is different from the previous work. The strength of this paper is that the experiments nicely demonstrate the effectiveness of the proposed approach. Also, the paper is well organized and well written. However, my concern is whether the training process will converge and achieves an equilibrium effectively. This paper will be more convincing if the authors could analyze and conduct experiments on this issue. I believe the contribution of this paper is important and I recommend this paper to be accepted.